# The Special and General Mechanism of Cyanobacterial Harmful Algal Blooms

**DOI:** 10.3390/microorganisms11040987

**Published:** 2023-04-10

**Authors:** Wenduo Cheng, Somin Hwang, Qisen Guo, Leyuan Qian, Weile Liu, Yang Yu, Li Liu, Yi Tao, Huansheng Cao

**Affiliations:** 1Division of Natural and Applied Sciences, Duke Kunshan University, 8 Duke Ave, Kunshan 215316, China; 2Guangdong Provincial Engineering Research Center for Urban Water Recycling and Environmental Safety, Tsinghua Shenzhen International Graduate School, Tsinghua University, Shenzhen 518055, China

**Keywords:** adaptive radiation, cyanobacterial harmful algal blooms (CyanoHABs), ecophysiology, mechanism, nutrient demand, purifying selection, water eutrophication

## Abstract

Cyanobacterial harmful algal blooms (CyanoHABs) are longstanding aquatic hazards worldwide, of which the mechanism is not yet fully understood, i.e., the process in which cyanobacteria establish dominance over coexisting algae in the same eutrophic waters. The dominance of CyanoHABs represents a deviation from their low abundance under conventional evolution in the oligotrophic state, which has been the case since the origin of cyanobacteria on early Earth. To piece together a comprehensive mechanism of CyanoHABs, we revisit the origin and adaptive radiation of cyanobacteria in oligotrophic Earth, demonstrating ubiquitous adaptive radiation enabled by corresponding biological functions under various oligotrophic conditions. Next, we summarize the biological functions (ecophysiology) which drive CyanoHABs and ecological evidence to synthesize a working mechanism at the population level (the special mechanism) for CyanoHABs: CyanoHABs are the consequence of the synergistic interaction between superior cyanobacterial ecophysiology and elevated nutrients. Interestingly, these biological functions are not a result of positive selection by water eutrophication, but an adaptation to a longstanding oligotrophic state as all the genes in cyanobacteria are under strong negative selection. Last, to address the relative dominance of cyanobacteria over coexisting algae, we postulate a “general” mechanism of CyanoHABs at the community level from an energy and matter perspective: cyanobacteria are simpler life forms and thus have lower per capita nutrient demand for growth than coexisting eukaryotic algae. We prove this by comparing cyanobacteria and eukaryotic algae in cell size and structure, genome size, size of genome-scale metabolic networks, cell content, and finally the golden standard—field studies with nutrient supplementation in the same waters. To sum up, the comprehensive mechanism of CyanoHABs comprises a necessary condition, which is the general mechanism, and a sufficient condition, which is the special mechanism. One prominent prediction based on this tentative comprehensive mechanism is that eukaryotic algal blooms will coexist with or replace CyanoHABs if eutrophication continues and goes over the threshold nutrient levels for eukaryotic algae. This two-fold comprehensive mechanism awaits further theoretic and experimental testing and provides an important guide to control blooms of all algal species.

## 1. Introduction

CyanoHABs are one of the most profound environmental hazards in modern human history in terms of their global geographical scale [1], longstanding duration (over a century) [2,3], and tremendous economic loss [4]. The mere fact that they are still intensifying and expanding under global climate change [1,5,6] attests to the fact CyanoHABs are still not fully understood, such that their ecology remains “complicated and confusing” [7]. To demystify CyanoHABs, they need to be examined from an evolutionary ecological perspective, considering all the main factors: bloom-forming cyanobacteria, other coexisting algal species, and eutrophic conditions. From an evolutionary perspective, all cyanobacteria as individual populations evolve in oligotrophic aquatic environments prior to water eutrophication and form blooms almost immediately in response to elevated nutrient loading [8,9]; ecologically, CyanoHABs are a consequence of uncontrolled growth of cyanobacteria in phytoplankton communities where coexisting algae do not form blooms at the same levels of nutrients [9,10,11]. In light of this analysis, the overarching question—of why cyanobacteria, not the coexisting algae, form blooms in the same eutrophic waters—can be broken down into two, one constituting the sufficient condition and the other the necessary condition: (1) how cyanobacteria form blooms in the eutrophic waters at the population level, and (2) why do coexisting algae not form blooms in the same eutrophic waters at the community level?

Both questions above boil down to the differential interplays between different life forms (cyanobacteria and coexisting algae) and environments in terms of energy and matter, leading to sustained and even saturated growth of cyanobacteria, but not of coexisting algae and thereby CyanoHABs. In this review, we first review the origin and evolution of cyanobacteria in oligotrophic conditions to provide the evolutionary ecological context for CyanoHABs, particularly in terms of the origin of relevant biological functions (ecophysiology). Next, we summarize the biological functions driving bloom formation, as a special mechanism of CyanoHABs at the population level. We then show that these functions are not a result of positive selection by water eutrophication, but of long adaptation to oligotrophic conditions prior to water eutrophication. Last, we propose a general mechanism of CyanoHABs at the community level to account for the fact that cyanobacteria but not coexisting algae form blooms in the same waters. 

## 2. Broad Niche Establishment by Cyanobacteria on Oligotrophic Earth

Extant cyanobacteria are found in nearly all habitats on this planet, from waters to hot springs to polar and arid climates [12], which are primarily oligotrophic. Given the quick or immediate bloom forming in response to water eutrophication [8,9], these functions reflect the already acquired ability to make the most of available nutrients. This brings one critical question: how have cyanobacteria acquired these functions? Revisiting the origin and evolution of cyanobacteria on early Earth provides important clues: these superior functions are shaped during their course of adaptive radiation.

Originating in an anoxic biosphere of early Earth, cyanobacteria are likely to evolve in a scarcity of macronutrients, such as phosphorus and nitrogen [13,14,15,16] and trace metals [17,18]. This humble beginning of life leads to primitive metabolic networks [19], which suggests that cyanobacteria could only obtain minimal energy and matter in both variety and quantity to support cell survival and division [20]. Arguably, their nutrient requirement for life is likely to be quite low. Besides nutrient deficiency, early cyanobacteria also faced various types of stresses which were “unforeseeable” for them. One type is the rising oxygen level they caused themselves through oxygenic photosynthesis, for which they developed various resisting strategies as seen in modern cyanobacteria [21,22]. Ultraviolet light is another early stress which has persisted until now, some cyanobacteria evolved against it by producing sunscreens [23]. These global stresses were also intertwined with landscape changes through geophysical processes (e.g., volcanism and climatic shifts). Here, we demonstrate the remarkable adaptive ability of cyanobacteria by showcasing their evolution of biological capability and morphology under the rising oxygen levels.

Since their origin, particularly after the Great Oxidation Event (GOE) (Figure 1), ancestral cyanobacteria quickly acquired versatile antioxidant capabilities against reactive oxygen species (ROSs), derived from O_2_ produced as a result of oxygenic photosynthesis. Different classes of superoxide dismutases (SODs) and other types of antioxidant enzymes evolved to deal with ROSs throughout their entire history [24]. Based on the local availability of metal cofactors (another prominent feature of adaptation), three types of SODs—NiSOD, CuZnSOD, and MnSOD/FeSOD—have been found in four types of aquatic environments (Figure 1A). Besides direct enzymatic removal of ROSs, cyanobacteria have also evolved other strategies, e.g., in photosynthetic reaction centers (RC) which split into two types as a response to the rising oxygen [25]. Another adaptation lies in the redundancy (i.e., heavy investment) of key functions. For example, in photosystem I, cyanobacteria have two electron transporters that evolved separately: copper-dependent plastocyanin (PC) and iron-dependent cytochrome c_6_ (Cytc_6_). Distinct in the primary sequence and tertiary structure of protein/amino acid, their expression is controlled by a transcription factor PetR (BlaI/CopY-family) and a BlaR-membrane protease (PetP), depending on the availability of the metal cofactors [26].

With the continued rise in oxygen level, morphological diversification was called upon to provide an extra dimension of protection. It has been confirmed that the evolution of multicellular morphotypes and the rate of morphological diversification coincide with the GOE onset [27]. Heterocyst became a specialized cell form for nitrogen fixation, in which the oxygen level is reduced to pre-GOE levels (Figure 1B). Nitrogen fixation is viewed as a leap forward in promoting marine primary production and contributed to increased O_2_ levels, coinciding with the rise in animals [28]. Similarly, cyanobacteria also obtained a specific photosynthetic structure for carbon fixation, carboxysome, in which CO_2_ is increased from 10–15 µM to 40 mM outside, which greatly improved the efficiency of the enzymatic fixation of carbon dioxide [29]. After the evolution of geological timescales (Figure 1C,D), modern cyanobacteria are among the most diverse prokaryotic phyla, with morphotypes ranging from unicellular to multicellular filamentous forms, including those able to terminally (i.e., irreversibly) differentiate in form and function [27,30].

Billions of years of adaptive evolution have allowed structural elaboration and functional diversification in coping with nutrient deficiency and environmental extremes and constraints [31]. This evolutionary divergence has enabled cyanobacteria to occupy all geographic habitats in modern Earth, including terrestrial and aquatic ecosystems, ranging from deserts to tropical rain forests, soils, and limestones, and from open oceans and brackish waters to freshwaters and hydrothermal vents (Figure 1E). This broad array of biological functions underlying the adaptive radiation in cyanobacteria is recorded in their genomes [32]. A large proportion of the genes in their genomes are associated with adaptation to the specific habitat they occupy, but not to any other habitats. Therefore, such genomic divergence leads to a pangenome with most genes being unique and only a small set of 323 genes being common among cyanobacteria [33]. These core genes are mainly involved in housekeeping functions, as opposed to interacting with the environment, e.g., ribosomal proteins, photosynthetic apparatus, ATP synthesis, chlorophyll biosynthesis, and the Calvin cycle. This small stable core and large variable shell of cyanobacterial genomes suggest most of the non-overlapping functions are specialized in adaptation to distinct habitats.

Based on the adaptation to the rising oxygen level that they produced themselves, paralleling the evolution of Earth and adaptation to various habitats (see discussion above), one can conclude that most of the functions in the CyanoPATH (metabolic pathways driving CyanoHABs) are also obtained through adaptive radiation. For example, one most prominent function is the ABC transport for phosphate PstSCAB, most cyanobacteria have two copies of operons and some top bloomers, such as *Microcystis aeruginosa*, have three operons. Again, CyanoPATH functions are the part of the entire function repertoire which sustained cyanobacteria through the harsh environment before the era of water eutrophication.

**Figure 1 microorganisms-11-00987-f001:**
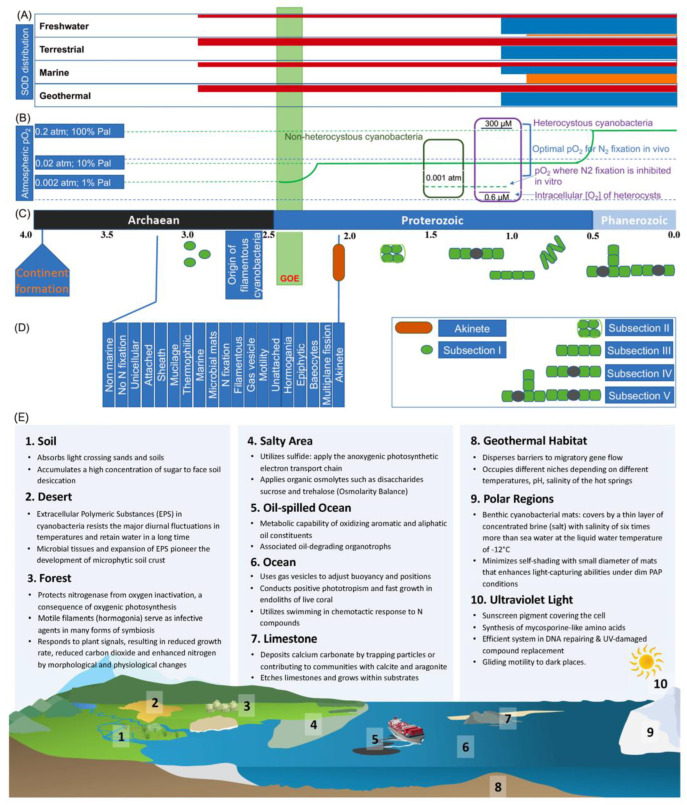
The evolutionary trajectory of cyanobacteria in the presence of oxygen and adaptive radiation: (**A**) The origin and distribution of SODs in different habitats. The timing of origin and habitat distribution of SOD genes among taxa is created based on Boden et al. [24]. Starting points of horizontal bars represent time of origin and color SOD types: CuZnSOD (blue), NiSOD (green), and Fe- and Mn-utilizing SODs (yellow). (**B**) The oxygen level in atmosphere and nitrogen fixation in heterocysts in cyanobacteria in the presence of elevated oxygen levels. The level of oxygen is based on [34]. (**C**,**D**) The temporal morphological diversification of cyanobacteria is based on Schirrmeistera et al. [27]. (**E**). Adaptive strategies of cyanobacteria in different habitats. Cyanobacterial adaptation strategies in nine environments, namely, soil, desert, forest, salty area, oil-spilled ocean, open ocean, limestone, geothermal water, and polar region. Cyanobacteria also cope with the damage caused by ultraviolet light.

## 3. The Special Mechanism of CyanoHABs

From an energy and matter perspective, the ubiquitous presence of CyanoHABs attests to the fact that bloom-forming cyanobacteria have metabolic machinery able to effectively assimilate external nutrients and convert them to biomass, aided by favorable weather or climate conditions [35,36]. That is to say that CyanoHABs are the outcomes of the synergistic interactions between the superior ecophysiology of cyanobacteria and eutrophic conditions. Indeed, both factors have been well established by field work, laboratory simulations, genomics, and metatranscriptomics. Specifically, the roles of nutrients, either trace elements or macronutrients, have been recognized as important drivers for CyanoHABs in a consensus and large survey [10,37]. Meanwhile, the metabolic pathways importing and converting available nutrients have been found to be more complete and have more copies of genes encoded in the genomes of bloom-forming cyanobacteria than non-blooming cyanobacteria [38,39] (Figure 2). These metabolic pathways and their genomes are now curated in a webserver CyanoPATH [39]. Most of the functions are peripheral and involved in nutrient uptake and precursor processing to feed central metabolism, and the rest are in the central metabolism converting metabolites for the synthesis of macromolecules in anabolism. Interestingly and expectedly, the gene expression levels of these pathways are correlated with their completeness [38]. To sum up, at the population level, the synergistic interactions between superior metabolic capabilities and elevated nutrients suffice as a working mechanism of CyanoHABs. We call this working mechanism a special mechanism, as opposed to the general mechanism proposed in Section 5.

## 4. Is there a Priori Adaptation to Eutrophic Water?

Given their global persistence for over a century [2,3], one may wonder whether there is prior adaptive evolution in cyanobacteria toward water eutrophication to form blooms, or whether CyanoHABs are simply ecological consequences of synergistic interactions between superior pre-equipped biological functions and elevated nutrients. We think the latter is more likely than the former. First, short-time whole-lake experiments in fertilization or nutrient reduction [40,41,42] have indisputably shown the causality of eutrophication for CyanoHABs on an ecological time scale, allowing no time for evolution. Second, all evolution of microbes is a passive, rather than proactive, response to environmental or genetic changes which are unforeseeable to them, and therefore they cannot proactively prepare themselves for water eutrophication. Third, molecular evolutionary analyses have shown that all genes, either the genes encoding CyanoPATH or the housekeeping genes or other genes, are not under positive selection by water eutrophication but are under strong purifying selection instead [43]. The dN/dS ratios (the ratio of divergence at nonsynonymous and synonymous sites (dN/dS) < 0.85 (median = 0.3)) for all homologous genes are similar between the genes in the pathways driving CyanoHABs and housekeeping functions. Fourth, there is also phylogenetic support for purifying: the strains of the same species from diverse geographic origins form the same clusters, while strains from the same origins form different clusters. Furthermore, the numbers of SNPs (single nucleotide polymorphisms) are found to be between 5 and 50, and interestingly, as the SNP number increases, the gene expression level decreases, which suggests that those with less SNPs (more conserved) are highly expressed (needed for their functions). These results suggest that these superior biological functions for CyanoHABs are acquired prior to water eutrophication.

## 5. The General Mechanism of CyanoHABs

In light of the special mechanism discussed above whereby CyanoHABs result from the synergistic interactions between elevated nutrients and the superior functions shaped during evolution, why it is cyanobacteria, not the coexisting eukaryotic algae, that form blooms in the same eutrophic waters? In what ways are cyanobacteria better than their co-living counterparts [44,45]? This question is so important that the population-centric special mechanism of CyanoHABs must be amended to accommodate the community-level dominance.

Here, we discuss this issue from a life form perspective. Compared to CyanoHABs, the coexisting algae cannot form blooms for two possible reasons. For one, they are not able to obtain enough nutrients for growth; for the other, they have higher demands for nutrients than the current levels can provide. In the discussion below, we focus on the latter and elucidate that cyanobacteria are simpler life forms than their co-living eukaryotic algae life forms and thus require less nutrients, as a result of their cell structure, size, genome size, per capita cell content, nutrient demand, and field manipulation.

### 5.1. Cell Structure and Size

Early deep origin and subsequent adaptive radiation (see Section 1) have made cyanobacteria the simplest form of life among algae. Additionally, cyanobacteria are the ancestor of eukaryotic algae, Chlorophyta, Rhodophyta, Euglenophyta, Cryptophyta, and Dinophyta commonly seen in fresh waters, e.g., in Lake Taihu (China) (Niu et al., 2011). Therefore, Cyanophyta, being a prokaryote, have the simplest cell structure among modern algae (Figure 3A): Cyanophyta does not contain a nucleus or membrane-bound organelles such as mitochondria, Golgi apparatus, and endoplasmic reticulum. Throughout the algal evolution, diverse eukaryotic algae have gained structural complexity and acquired plastids via endosymbiotic events on their common prokaryotic ancestor, Cyanophyta (Figure 3B). In parallel to their simple structure, cyanobacteria have the smallest size among all algae groups [46]. One example is provided with lab cultures: cyanobacterium *M. aeruginosa* and chlorophyte *Cholorella vulgaris* are 3.30 ± 0.91 μm and 5.32 ± 1.27 μm in diameter, respectively [47].

### 5.2. Genome Size

Consistent with the structural simplicity, the genome sizes of cyanobacteria are the smallest compared to those of eukaryotic algae, which are smaller by one or two orders of magnitude (Figure 4A). Most genomes of the eukaryotic algae are about the same size, of around 100 Mb base pairs, except those of Phaeophyceae and Dinophyceae being one more order of magnitude larger. Cyanobacteria usually have around 5000 genes [48,49], while eukaryotic genomes have at least twice as many [50,51]. These differences in genome size and gene count reflect the difference in cell structure and functions.

### 5.3. Genome-Scale Metabolic Networks (GSMs)

Cyanobacteria have half the number of the genes encoded in most eukaryotic genomes. For some species, GSMs have been reconstructed based on the gene–protein/enzyme-reaction rule and experimental measurement of the macromolecule components (Table 1). In terms of the number of reactions and metabolites, the GSMs of cyanobacteria have, on average, 780 metabolites and 820 reactions, which are only less than half those of the eukaryotic GSMs. As the reactions are involved in catabolism, core metabolism, and anabolism [52], these smaller GSMs in cyanobacteria make it clear that less nutrient input is needed than their more complex eukaryotic counterparts for life maintenance and growth. Therefore, as nutrient levels increase from the oligotrophic states during the early process of eutrophication, cyanobacteria are the first group that can be saturated for growth.

Besides the size of GSMs, the metabolic rate (time^−1^) is known to often scale with cell size with a volume scaling exponent *b* = −0.25 [53,54].

**Table 1 microorganisms-11-00987-t001:** Comparison of genome-scale metabolic networks between cyanobacteria and eukaryotic algae.

	Organism	Genes	Model Genes	Metabolites	Reactions	Reference
Cyanophyta	*Arthrospira platensis*	6631	620	673	746	[55]
	*Cyanothece* sp. ATCC 51142	4912	806	651	667	[56]
	*Spirulina platensis* C1	6153	692	837	875	[57]
	*Synechococcus elongatus*	2723	785	768	850	[58]
	*Synechococcus* sp. PCC 7002	3179	821	777	792	[59]
	*Synechocystis* sp. PCC6803	3221	678	795	863	[60]
	*Synechocystis* sp. PCC6803	3221	669	790	882	[61]
Chlorophyta	*Chlamydomonas reinhardtii*	16,709	866	1862	1725	[62]
	*Chlamydomonas reinhardtii*	16,709	1080	1068	2190	[63]
	*Chlorella variabilis*	9879	526	1236	1455	[64]
	*Chlorella vulgaris*	7100	843	1770	2286	[65]
Bacillariophyta	*Phaeodactylum tricornutum* CCAP 1055/1	12,280	1027	2172	4456	[66]
	*Schizochytrium limacinum* SR21	63,000	1170	1659	1769	[67]

### 5.4. Per Capita Cell Content

Following the same line of life form complexity, we examined the cell content per cell between cyanobacteria and eukaryotic algae. It is clear that different species of cyanobacteria have lower cell content than those of other algae, using the data from the literature [68,69] (Wilcoxon Test, *p* < 0.01; Figure 4B). Interestingly, species of cyanobacteria or non-cyanobacteria show a similar trend of within-group variation in per capita cell content (Figure 4C). Here, we provide a specific example of lab cultures. Dry weight of individual *Cholorella vulgaris* cells at exponential phase are 1.70 ± 0.07 × 10^−11^ g/cell, and those of *M. aeruginosa* average 6.38 ± 0.05 × 10^−12^ g/cell, which is about the same as the green alga [47].

**Figure 4 microorganisms-11-00987-f004:**
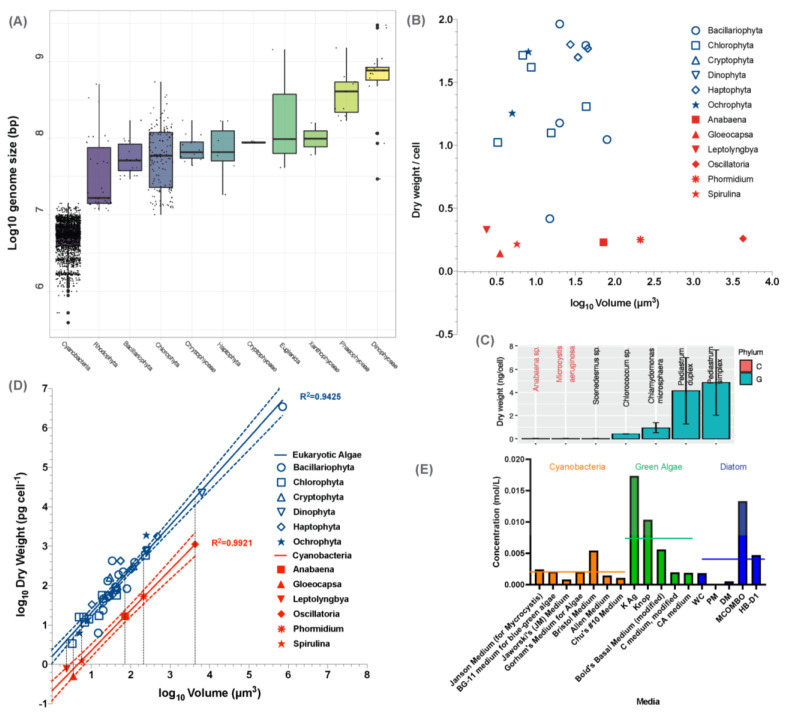
Comparison between cyanobacteria and eukaryotic algae in select key traits: (**A**) Distributions of log-transformed sequence length in eleven groups of algae. Summary statistics of 2240 genome assemblies were downloaded from NCBI Datasets: 1977 in cyanobacteria, 38 in Rhodophyta, 26 in Bacillariophyta, 156 in Chlorophyta, 14 in Chrysophyceae, 7 in Haptophyta, 3 in Cryptophyceae, 3 in Cryptophyceae, 3 in Euglenida, 2 in Xanthophyceae, 11 in Phaeophyceae, and 20 in Dinophyceae, respectively. Each dot in the boxplot represents the log-transformed total sequence length of one assembly. (**B**) Distributions of dry weight per cell biovolume in Eukaryotic algae and cyanobacteria from the literature [68,69]. Each dot plot represents the average values of dry weight per unit biovolume and cell dry weights within a class. Blue symbols represent eukaryotic algae cells and red symbols represent cyanobacteria. (**C**) Relationship between cell volume (log_10_*V*, µm^3^) and dry weight (log_10_*DW*, pg cell^−1^). Each dot plot represents the average values of cell volumes and cell dry weights within a class. Linear regressions were applied for eukaryotic algae (blue) and cyanobacteria (red) with 95% confidence intervals. (**D**) Lower nutrient requirements by cyanobacteria than eukaryotic algae in aquatic environment. Among seven selected media, the recipes of mAC, MJ, VTAC, and CHU-11 (left panel) are used for culturing green algae and the recipes of M11, BG11, and CT (right panel) are used for culturing cyanobacteria. (**E**): Lower nutrient requirements by cyanobacteria than eukaryotic algae in aquatic environment. Among seven selected media, the recipes of mAC, MJ, VTAC, and CHU-11 (left panel) are used for culturing green algae and the recipes of M11, BG11 and CT (right panel) are used for culturing cyanobacteria.

### 5.5. Concentration of Synthetic Media

To further support the point that cyanobacteria are simpler life forms and require less energy and mass input, we examined the recipe of the commonly used media optimized for growing different groups of algae. As shown in Figure 4D, the media for culturing cyanobacteria generally have lower total concentrations of ions than those for green algae or diatom, although the statistic test, Welch’s test, produced a high *p* value (*p* = 0.1) for the selected media.

### 5.6. Golden Test of Nutrient Demand: Whole-Lake Nutrient Manipulation

To test the hypothesis that cyanobacteria have lower nutrient requirements for maximum growth than co-living eukaryotic algae, we summarized the results, which directly compared the growth of cyanobacteria with eukaryotic algae in the field, laboratory, and cosmos by supplying external nutrients (Table 2). Five case studies we discovered all support our hypothesis: low nutrients (TP, TN, DIN, urea-N, Ca, etc.) lead to dominance of cyanobacteria and high nutrients produce a dominance of green algae and even brown algae.

In a large study of 210 Danish lakes, heterocystous cyanobacteria were in dominance when the total phosphorus (TP) was at low levels (<0.25 mg P∙L^−1^) and non-heterocystous cyanobacteria were in dominance when TP was at intermediate levels (0.25–0.8 mg P∙L^−1^), while chlorophytes often dominated when TP was high (>1 mg P∙L^−1^) [70]. In another whole-lake study conducted in Wascana Lake (Canada) [71], different levels (0, 1, 3, 8, and 18 mg N L^−1^) of urea added into the surface water every seven days (day 0, day 7, and day 14) yielded compositional shifts in the phytoplankton community—after 21 days, colonial cyanobacteria dominated when the N level was moderate (≤8 mg N L^−1^), and chlorophyte dominated when the N level was high (>8 mg N L^−1^). A third support comes as a nutrient-supplemented mesocosm in the Archipelago Sea (Table 2). Two ratios of N:P supplement solution, 1N (1.7 μg L^−1^):1P (1.7 μg L^−1^) and 7N (12 μg L^−1^):1P (1.7 μg L^−1^), were applied to the mesocosm at low initial concentrations of N (mean = 2.7 μg L^−1^) and P (mean = 2.8 μg L^−1^) every day over twenty days. At the onset, heterocystous cyanobacteria made up 70% to 80% of the total phytoplankton community. Compared to 1N:1P treatment, 7N:1P led to the most dramatic decline in the percentage of the heterocystous cyanobacteria and the chlorophyte dominance with the flourishing of *Dictyosphaerium subsolitarium*, *Monoraphidium contortum*, and *Oocystis* spp. Besides N and P, the need for calcium is also considered. Lab-based calcium enrichment experiment (Wu and Kow, 2010) showed that both the cell number and the cell column of chlorophytes elevated with the increase in calcium level, in contrast to those of cyanobacteria. More examples of nutrient supplementation to shift dominance from cyanobacteria to other algae are provided in Table 2.

To sum up, cyanobacteria are simpler life forms than co-living eukaryotic algae, in cell size, cell structure, genome size, GSMs, and cell content. Simple structure and small size provide them with higher metabolic rates, leading to higher abundance [46], which have been proved by field nutrient supplementation. Now, it is clear that complex eukaryotic algae would form blooms if nutrients were high enough.

**Table 2 microorganisms-11-00987-t002:** Case studies of nutrient manipulation on phytoplankton in situ, in the mesocosm study, and in laboratory conditions showing lower nutrient requirements of cyanobacteria than eukaryotic algae. “C” represents cyanobacteria; “G” represents green algae. DIN: dissolved inorganic nitrogen; SRP: soluble reactive phosphorus; TN: total nitrogen; and TP: total phosphorus.

Study	Nutrient Variation	Dominant Species	Dominance
Wascana Lake, Canada Mesocosm Study [71]	1–8 mg Urea-N L^−1^	Colonial cyanobacteria	C
	>8 mg Urea-N L^−1^	Chlorophytes	G
210 Danish lakes in situ Study [70]	0.25–0.8 mg TP L^−1^	Non-heterocystous cyanobacteria	C
	>1.0 mg TP L^−1^	Chlorophytes	G
Lake Tai, China Laboratory and in situ study [72]	3 mg/L (TN) and 0.2 mg/L (TP)	*M. aeruginosa*	C
	10, 15, or 20 mg/L (TN), and 1 mg/L (TP)	*Scenedesmus quadricauda*	G
Archipelago Sea Mesocosm Study [73]	1N:1P 1.7 μg L^−1^:1.7 μg L^−1^	*Aphanizomenon* sp., *Nodularia spumigena*, *Anabaena* spp., *Synechococcus* spp.	C
	7N:1P 12 μg L^−1^:1.7 μg L^−1^	*Dictyosphaerium subsolitarium*, *Monoraphidium contortum*, and *Monoraphidium contortum*, *Oocystis* spp.	G
Taipei Feitsui Reservoir Laboratory and in situ study [74]	7–9 mg Ca^2+^ L^−1^	*Microcystis* spp. and *Aphanocapsa delicatissima*.	C
	9, 11, 13, 15, 18 mg Ca^2+^ L^−1^	*Eutetramorus*, *Coelastrum*, *Coenocystis*, and *Dictyosphaerium*	G
Laboratory Study [75]	32 μM (NO_3_) after 48 h; 32 μM NH_4_^+^ after 48 h	*Synechococcus* 6301	C
	32 μM NO_3_ after 25 h; 32 μM NH_4_ after 18 h	*Scenedesmus* sp.	G
Vila Lake, Central Portugal in situ study [76]	0.66–2.42 mg L (DIN); 0–0.24 mg L (SRP)	*Aphanizomenon flos-aquae*, *Ch* *roococcus limneticus*, *M. aeruginosa* and *Pseudanabaena* sp.	C
	0.7–3.3 mg L (DIN); 0.19–0.93 mg L (SRP)	*Coelastrum reticulatum var. reticulatum*, *Kirchneriella lunaris*, *Monoraphidium contortum*, *Scenedesmus acuminatus var. acuminatus* and *Pediastrum boryanum var. boryanum*	G

## 6. Synthesis and Outlook

We propose a synthesis of CyanoPATH as a result of the synergistic interactions between cyanobacterial superior functions and elevated nutrients. In this evolutionary ecological perspective, we provided evidence that these functions are shaped during the long course of adaptive radiation, but not toward water eutrophication. More importantly, we place CyanoHABs in the community context and show that they are simpler life forms than co-living eukaryotic algae in terms of cell size, cell structure, genome size, per capita cell content, metabolic network, and nutrient demand. At last, simpler cyanobacteria having low nutrient requirements are proved by various golden tests, including field studies of nutrient supplement. This community-centric mechanism of CyanoHABs provides a framework to study them in a wholistic perspective, considering both the differences in energy and matter requirements of various algal cell machineries and the environment dictating their dynamics in relation to each other.

Considering the multiple factors in the mechanism, further empirical tests and theoretic explorations are needed for a better synthesis. Two types of research should be given priority: (1) Comparative studies between cyanobacteria and co-living algae on nutrient requirements and growth in lab and field settings, and their interactions with varying levels of environmental factors. Direct comparison between CyanoHABs with blooms of other algae in different waters will also be useful but should be fully assessed for other confounding factors which may prohibit cyanobacterial growth; (2) The integrated systems biology assessment of the relative roles in promoting growth in cyanobacteria. The latter can be best achieved by genome-scale metabolic networks.

In application, our special and general mechanism highlights the need for cutting nutrient discharge into water bodies to control these blooms. If nutrients uncontrollably increase, our dual-level mechanism predicts that CyanoHABs will be replaced by blooms of other algal species with higher nutrient and energy requirements, e.g., green algae.

## Figures and Tables

**Figure 2 microorganisms-11-00987-f002:**
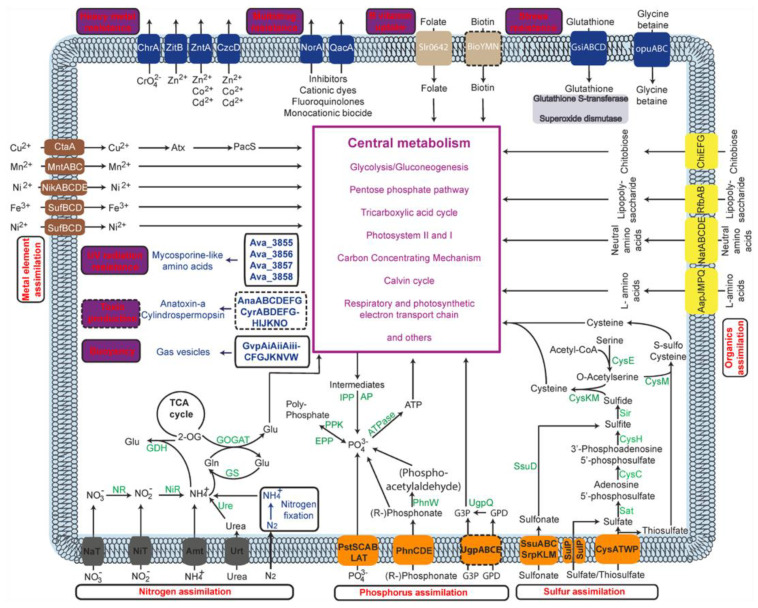
The core and query pathways in *Aphanizomenon flos-aquae* NIES-81. The pathways labeled with dashed borders are not complete (due to the absence of required components) in this strain but may be complete in others.

**Figure 3 microorganisms-11-00987-f003:**
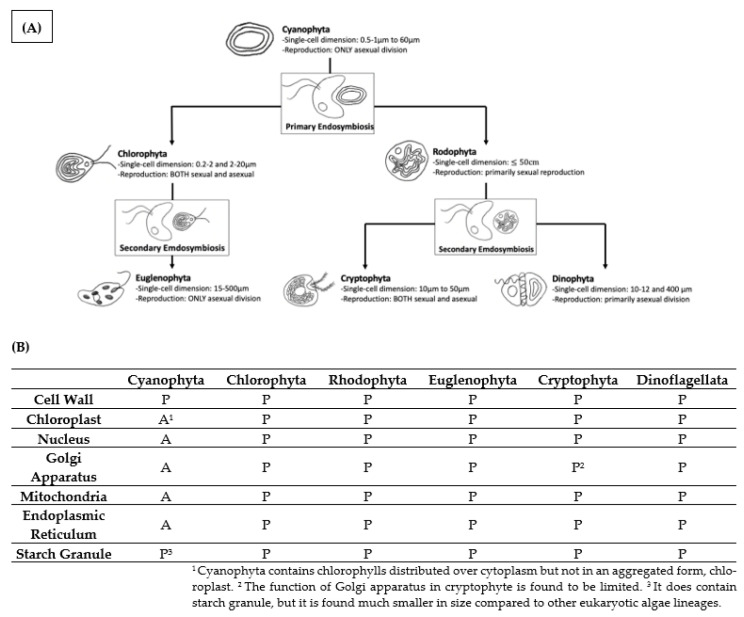
Cell structure of cyanobacteria and eukaryotic algae commonly seen in fresh waters and their evolutionary relationship. A: absent; P: present. (**A**) The major eukaryotic groups have risen from a common prokaryotic ancestor, Cyanophyta, via endosymbiotic events. Through the primary endosymbiosis, Chlorophyta and Rhodophyta acquired primary plastids, membrane-bound organelles usually found in eukaryotic organisms. Chlorophyta was preyed upon by a second eukaryotic cell, resulting in Euglenophyta acquiring their plastids. Similarly, Cryptophyta and Dinophyta have arisen as a result of secondary endosymbiosis on Rhodophyta. (**B**) Comparison of cellular components in six common major algal phyla. Cyanophyta shows the absence of nucleus, chloroplast, and mitochondria compared to other eukaryotic algae lineages. Cyanophyta, being the common prokaryotic ancestor, do not contain membrane-bound organelles.

## Data Availability

Not applicable.

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
