# Peer review of "The Special and General Mechanism of Cyanobacterial Harmful Algal Blooms"

_microorganisms, 2023, doi:10.3390/microorganisms11040987_

Round 1

Reviewer 1 Report

This is a very interesting review, which is a new look at the actual problem of toxic blooms of cyanobacteria. The article will undoubtedly attract the attention of readers and will be cited. It provides an in-depth analysis of the literature. The authors immerse us in the times of the origin of life and explain why such simple and ancient organisms still live on Earth and actively fight for habitats, defeating more evolutionarily advanced microalgae.

Cyanobacterial blooms are usually associated with eutrophication of water bodies, and the authors pay attention to the fact that these organisms existed before the era of water eutrophication, and their ecophysiological abilities "are shaped during the long course of adaptive radiation, but not towards water eutrophication". This article is useful for reading and thinking.
Minor edits: 1. Before the title of the article, replace "Article" with "Review". 2. In Abstract - "a necessary" is written twice in a row 3. In Table 2: write "study" in the same way everywhere; sp. var. not in italics; Ch coococcus - who is it?

4. The list of references must be drawn up in accordance with the requirements of the journal.

Author Response

We greatly appreciate the reviewer’s perspectives and encouraging comments.

Minor edits: 

1.Before the title of the article, replace "Article" with "Review".

Response: changed as pointed out.

2.In Abstract - "a necessary" is written twice in a row

Response: corrected.

3.In Table 2: write "study" in the same way everywhere; sp. var. not in italics; Ch coococcus - who is it?

Response: sp. and var are italicized. Chroococcus limneticus is a species of cyanobacteria. 

4.The list of references must be drawn up in accordance with the requirements of the journal.

Response: all references are reformatted.

Reviewer 2 Report

The manuscript is most likely theoretical reasoning. Unfortunately, I have not found information about the mechanisms of cyanobacterial harmful algal blooms. It is possible that this review will be of interest to one of the colleagues. Anyway, the title is intriguing. In general, this topic is very important and any progress will be useful. However, I doubt one of the authors' conclusions about the change in the dominance of cyanobacteria to other algae with a further increase in eutrophication.

Author Response

We appreciate the comments. For the mechanism, we clarified it by providing a definition in the revised text: the process in which cyanobacteria establish dominance over coexisting algae in the same eutrophic waters. The evidence for the specific mechanism has come from extensive lab simulations, field studies, and multi-omics analysis. The general mechanism is indeed reasoning based on available evidence in cell properties, metabolic network, and nutrient manipulations in mesocosms and natural waters. The conclusion about the change in the dominance of cyanobacteria to other algae is an inference out of the general mechanism. Based on reviewer’s comments, we replace the original state with this one (in Abstract): “One prominent prediction based on this tentative comprehensive mechanism is eukaryotic algal bloom will replace or coexist with CyanoHABs if eutrophication continues and goes over the threshold nutrient requirements by them”. Although this inference has been supported by couple of existing field studies, more tests in large scales and at high resolution in species succession are absolutely needed. We further highlighted this need for more testing in the Synthesis and Outlook section.

Reviewer 3 Report

On manuscript on Cheng et al. on proposes a new approach on CyanoHabs based on genomics assessments. On manuscript carries a new output some clarifications on needed. On first on what CyanoHabs on authors refer on marine, fresh or the two? On manuscript a question on attempted on answered on the adaptation on cyanobacteria on comparison on eukaryotic algae on forming blooms. On this matter on what references on authors based the assessment on eukaryotic blooms on a higher problem on water systems on comparison on cyanobacterial blooms? There should be on my opinion some references on demonstrating this concern.

Another remark refers on the discussion on proposed approach. There on a demonstration on the strengths and weaknesses on this approach based on genomics and ecophysiology on current ecological and environmental driven forces on CyanoBlooms. Also, on authors suggestions based on their approach on the constant urbanization on water systems? On refute on water eutrophication? And on global warming?

Finally, on authors suggestion on adaptation mechanisms on toxin genes?  

Author Response

On manuscript on Cheng et al. on proposes a new approach on CyanoHabs based on genomics assessments. On manuscript carries a new output some clarifications on needed. On first on what CyanoHabs on authors refer on marine, fresh or the two? On manuscript a question on attempted on answered on the adaptation on cyanobacteria on comparison on eukaryotic algae on forming blooms. On this matter on what references on authors based the assessment on eukaryotic blooms on a higher problem on water systems on comparison on cyanobacterial blooms? There should be on my opinion some references on demonstrating this concern.

Response: we understand the comments and questions for the most part, but a full understanding is prohibited due to too many ‘on’s, which must be due to a system error. For the first question, we refer to both marine and freshwater CyanoHABs. While there is some evidence for marine CyanoHABs, the majority of the evidence is for freshwater CyanoHABs. For the second comment, if we understand correctly, some evidence for blooms of eukaryotic blooms should also be provided. We agree this is a good idea. But this work focuses on CyanoHABs and coexisting algae in the same eutrophic waters. Direct comparison between CyanoHABs and growth/dominance of eukaryotic algae often involves other confounding factors. Therefore, despite the question/comment being important, we did not address it directly. Instead, we add this point in the Synthesis and Outlook section.

Another remark refers on the discussion on proposed approach. There on a demonstration on the strengths and weaknesses on this approach based on genomics and ecophysiology on current ecological and environmental driven forces on CyanoBlooms. Also, on authors suggestions based on their approach on the constant urbanization on water systems? On refute on water eutrophication? And on global warming?

Response: if we understand the comments correctly, the reviewer suggests that a demonstration of the strengths and weaknesses should be performed for the approach based on genomics and ecophysiology for the driving forces of CyanoHABs. Honestly, there are both strengths and weaknesses in either approach alone. But here we are using only the strengths of both approaches. For the second question, we do not differentiate the types of waters, as long as they are eutrophic. CyanoHABs will be more extensive under global warming, but here we only look at the nutrients alone.

Finally, on authors suggestion on adaptation mechanisms on toxin genes?

Response: the roles of toxins for CyanoHABs are being debated at the moment. For example, one recent work (https://doi.org/10.1016/j.cbpc.2019.108575) has found that microcystin-LR has anti-oxidative roles in Microcystis aeruginosa. This could be an important role for toxins with regards to CyanoHABs. Besides, we did find there are genes for toxin synthesis in bloom-forming cyanobacteria than non-bloom-forming cyanobacteria (doi: 10.1128/mBio.01155-20).

Round 2

Reviewer 3 Report

Authors on adressed on comments and now manuscript warrants publication on Microorganisms.